# A Small RO and MCDI Coupled Seawater Desalination Plant and Its Performance Simulation Analysis and Optimization

**Shouguang Yao \* and Mengting Ji**

School of Energy and Power, Jiangsu University of Science and Technology, Zhenjiang 212000, China; ntmengtingji@163.com

\* Correspondence: zjyaosg@just.edu.cn

**Abstract:** To solve the problems of high specific energy consumption and excessive harmful ions in the water production of a small reverse osmosis (RO) plant, a desalination system coupling RO and membrane capacitive deionization (MCDI) is proposed in this study. Aiming at producing two cubic meters per day of fresh water with a salt concentration of less than 280 mg L$^{-1}$, parameter matching optimization was carried out on two desalination system schemes of one-stage two-section RO and one-stage three-section RO coupled with MCDI. The results were compared with the parameter matching optimization results of the one-stage one-section RO and the one-stage two-section pure RO desalination system. The results show that compared with the pure RO desalination mode, the seawater desalination mode coupled with RO and MCDI reduces the specific energy consumption under the same effluent salt concentration. Moreover, it decreases the feed water pressure in front of the RO membrane, which can reduce the standard of high-pressure pump in a small seawater desalination plant. The energy consumption of the one-stage three-section RO and MCDI coupling system is lower than that of the one- stage two-section RO and MCDI coupling system, and the feed water pressure is also lower.

**Keywords:** coupled seawater desalination; membrane capacitive deionization; reverse osmosis; simulation analysis and optimization

---

## 1. Introduction

Given the limited supply of fresh water on earth, seawater desalination is an effective way to produce this valuable resource. Small-scale seawater desalination technology has incomparable advantages, such as small footprint, easy transportation and simple and flexible installation [1]. Ordinary users who lack access to fresh water, such as those in long-distance ships, can obtain fresh water through this technology. In traditional seawater desalination, reverse osmosis (RO) desalination has low energy consumption and compact structure, and the RO desalination process is used by approximately 80% of the total desalination plants worldwide [2]. Although one-stage RO desalination can achieve the salt concentration standard for direct drinking water, the produced water is acidic and retains excessive harmful elements, such as bromine ion [3]. If a two-stage RO system is adopted, only part of the water quality problems can be solved, and the energy consumption and cost will greatly increase. In particular, the existing small-sized RO seawater desalination plant needs small water flow because of its small daily water production. To increase the water recovery of the system and reduce the specific energy consumption, it not only needs larger feed water pressure, which greatly improves the high-pressure pump standard, but it also needs to maintain the specific energy consumption as high as 10–20 kWh m$^{-3}$ [4] without an energy recovery device. Membrane capacitive deionization

(MCDI) technology, as a new way of seawater desalination, is more effective than the traditional desalination technology because of the unique method that MCDI technology uses in removing salt ions from water [5]. It has lower energy consumption in theory, has no secondary pollution, and can effectively remove other ions that are harmful to the human body and metal pipes. However, as far as the current technology is concerned, it is not the dominant method used in the desalination of high-salinity raw water. According to one report [6], the specific energy consumption of CDI is lower than that of RO in certain cases. According to the experimental verification of Porada et al. [5], MCDI has an evident advantage in energy consumption when it deals with the solution with a concentration below 30 mmol $L^{-1}$ compared with the RO method.

Few studies have been conducted on the coupling desalination method between RO and MCDI. In 2009, Lee et al. [7] used the CDI process to recover the wastewater produced by the second-stage RO (RO wastewater TDS is $1276 \pm 166$ mg $L^{-1}$) for domestic RO water purification equipment. This method is not only able to recover at least 85% of the RO rejected from the water reclamation facility, but it can also remove more than 88% and 87% of TDS and ion, respectively. Thus, the product quality of CDI is equal to or better than that of the RO influent.

In 2013, Jande et al. [8] proposed the RO and constant voltage CDI mixing system (RO–CVOCD) to produce ultrapure water and drinking water. In this study, the optimal operating parameters have been selected for different feed seawater concentrations to make RO desalination energy as low as possible. Then, the RO-produced water enters MCDI desalination to obtain ultrapure water and qualified drinking water. However, the two desalination methods are only mechanically combined and not considered for parameter matching. For example, when the feed water concentration is 35,000 ppm, the feed water flow rate is 100 mL $s^{-1}$ (8.64 $m^3$ $d^{-1}$), and the feed water pressure is 78 bar, thus leading to a high feed water pressure of the high-pressure pump.

In 2014, Minhas et al. [9] proposed to use the desalination method of RO and constant current MCDI (RO–CCOCD) to produce ultrapure water and drinking water on the basis of reference [8]. The results show that under the same design parameters of an MCDI device, the specific energy consumptions of the RO–CCOCD system and the RO–CVOCD system are similar. The amount of pure water produced by CCOCD is larger, but the concentration of pure water is slightly higher. However, the system is still a mechanical combination of the two methods, and the requirements of a high-pressure pump are very high.

In 2013, Minhas et al. [10] proposed a scheme to treat brackish water by using the mixed desalination method of RO and CDI. RO desalted the water first. Then, the resulting wastewater (1686 ppm) underwent CDI for secondary desalination. Afterward, the RO permeate water was mixed with CDI desalting water to obtain the final product water. The results show that the specific energy consumption of the RO+CDI system is not only lower than that of the RO+RO system, but the water recovery rate is also higher.

In 2018, Dorji et al. [11] suggested using MCDI instead of a secondary-stage RO device to treat bromide in a one-stage RO permeate. An experimental-scale MCDI device is used to treat permeate water of different qualities. The results show that under the same feed water concentration and feed water flow rate, the average energy consumption of the secondary-stage RO is 40% higher than that of MCDI for the average feed TDS of 300 mg/L. Furthermore, MCDI can effectively remove bromide for the TDS range, which is normally associated with the first-stage RO permeate.

In 2019, Choi et al. [12] proposed to use the RO-MCDI-RED (reverse electrodialysis) mixed seawater desalination system to improve energy efficiency. For the same one-stage RO-produced water quality, the comparison between the experimental results of RO+MCDI and the simulation results of RO+RO proves that the specific energy consumption of the combined RO+MCDI is lower than that of the combined RO+RO. However, no corresponding optimized design is proposed. Instead, the two methods are mechanically combined when the RED device is not used. MCDI flushing liquid is directly mixed with RO wastewater, thus increasing the specific energy consumption of the system.

In summary, the combination of RO and CDI desalination methods proposed in the existing literature only mechanically couples the two together, and the studies do not take into account the stringent requirements for small water flow rate and high premembrane pressure of the one-stage RO of the small-scale seawater desalination system. Moreover, the optimization scheme for coupling the two parts of the system is not considered, thus resulting in the need for high-standard, high-pressure pumps and excessive energy consumption.

In this study, a seawater desalination system with the coupling of RO and MCDI is proposed. Then, the lowest energy consumption of the system is taken as the optimization goal. The water production salt concentration is lower than 280 mg $L^{-1}$, and the daily water production is more than two cubic meters as the constraint conditions. The different coupling design schemes of the RO and MCDI coupling seawater desalination system are compared and analyzed. Then, the parameters are optimized.

## 2. System Design

The system of seawater desalination coupled with RO and MCDI proposed in this study is shown in Figure 1. The coupling design scheme provided in figure is a schematic diagram of a waterway system with the coupling of one-stage two-section RO and MCDI.

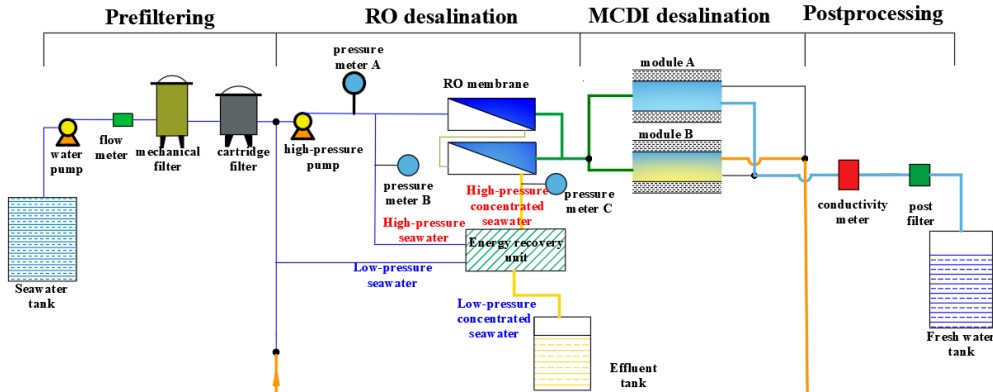

**Figure 1.** Schematic diagram of waterway system with coupling of one-stage two-section reverse osmosis (RO) and membrane capacitive deionization (MCDI).

The process of the system is as follows:

(1) RO primary desalination waterway. The feed seawater after pretreatment is divided into two lines. The first line enters the RO module through the high-pressure pump. The second line flows through the high-pressure energy recovery device [13]. Then, the line is pressurized by high-pressure concentrated seawater. After achieving the feed water pressure, the seawater is collected with the first line of high-pressure seawater into the RO module for RO desalination.

The schematic diagram of the RO energy recovery unit is shown in Figure 2. In the RO energy recovery unit, the high-pressure concentrated seawater in front of the RO membrane enters concentrated water chamber A or B to push the piston to move. The area ratio of the seawater chamber to the concentrated water chamber pressurizes the seawater to reach the feed pressure required by the RO membrane assembly. The seawater pressurized by the energy recovery device is discharged through seawater chamber A or B, which is collected together with the high-pressure seawater at the outlet of the high-pressure pump and flows into the RO membrane module.

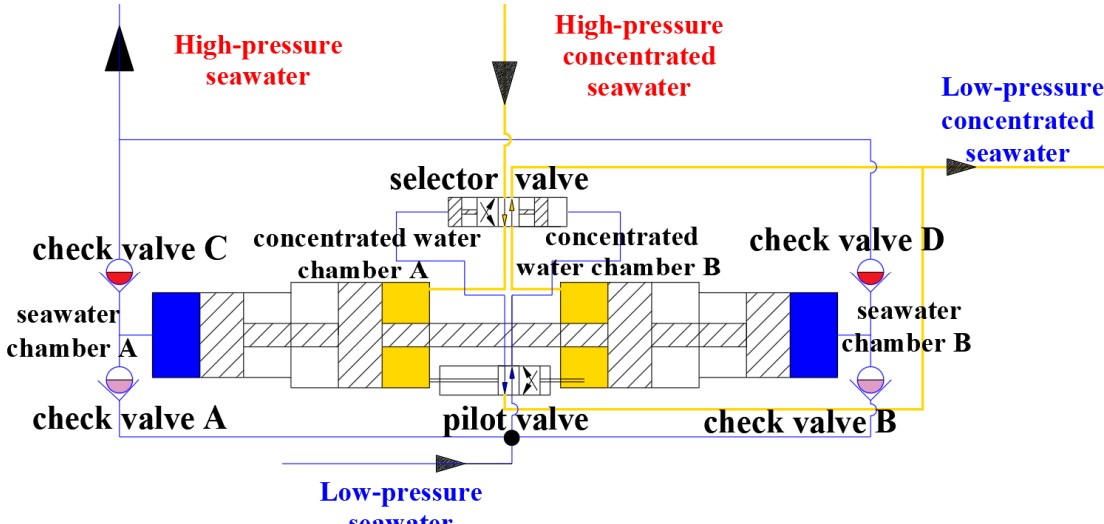

**Figure 2.** Schematic diagram of RO energy recovery unit.

(2) Desalination waterway of MCDI module. The outlet of the RO module is connected with the inlet of the MCDI module. The MCDI module works alternately by two components, which carry out adsorption and desorption at the same time to ensure the continuity of the waterway. For example, when component A enters the adsorption operation, component B enters the desorption operation. First, the pump is stopped, and the power is turned off for a certain period. Then, the power is reverse charged for a period of time. Finally, the pump is turned on and rinsed with RO osmotic water to complete the desorption process. The washed solution is returned to the front of the high-pressure pump of the RO module. Then, it enters the system with the feed water to be desalted again. No wastewater is generated at the MCDI module.

## 3. Mathematic Model

### 3.1. Desalination Model

On the basis of the Kimura–Sourirajan [14] model and thin membrane theory [15], the mathematical model of the RO membrane is established to describe the transport and concentration polarization of this membrane. The water flux passing through membrane $j_w$ is expressed as

$$j_w = A_w(\Delta P - \Delta \pi) \tag{1}$$

$$\Delta P = \frac{P_f + P_c}{2} - P_p \tag{2}$$

$$\Delta \pi = \pi_m - \pi_p \tag{3}$$

where $A_w$ is the permeation constant of pure water. $\Delta P$ and $\Delta \pi$ are the difference of static pressure and osmotic pressure across the membrane. $P_f$, $P_c$ and $P_p$ are the feed water pressure, the concentrated water pressure and the production water pressure of RO, respectively. $\pi_m$ and $\pi_p$ are the osmotic pressure on the membrane wall on the feed side and the osmotic pressure of permeate water, respectively.

The formula for calculating osmotic pressure is [16].

if $C < 20,000$ mg L$^{-1}$,

$$\pi = \frac{C(T + 320)}{491,000} \times 10^5 \tag{4}$$

if $C > 20,000$ mg L$^{-1}$,

$$\pi = \left( \frac{0.0117C - 34}{14.23} \right) \frac{(T + 320)}{345} \tag{5}$$

where $C$ is the concentration and $T$ is the temperature in Celsius.

The salt flux through the membrane is

$$j_s = B_s\left(C_m - C_p\right) \tag{6}$$

where $B_s$ is the solute transport parameter. $C_m$ is the concentration of the membrane wall on the influent side. $C_p$ is the permeate water concentration produced by one-stage RO.

The concentration and flow rate of the water produced by one-stage RO can be expressed as

$$C_p = \frac{j_s}{j_w} \tag{7}$$

$$Q_p = A j_w \tag{8}$$

where $A$ is the effective area of the membrane.

The conservation formulas of the solvent volume and the solute mass are as follows, respectively:

$$Q_f = Q_c + Q_p \tag{9}$$

$$Q_f C_f = Q_c C_c + Q_p C_p \tag{10}$$

where $Q_f$ is the feed water flow rate and $Q_c$ is the concentrated water flow rate.

The trapped solute forms a concentration polarization layer on the surface of the membrane, which leads to the decrease of membrane separation ability. The concentration polarization factor ($\phi$) can be expressed as

$$\phi = \frac{C_m - C_p}{C_B - C_p} \tag{11}$$

$$C_B = \frac{C_f + C_c}{2} \tag{12}$$

where $C_B$ is the average salt concentration on the feed side.

Concentration polarization factor is a crucial parameter in simulation analysis. The operation parameters that affect the concentration polarization factor are feed water pressure and feed water flow rate. The rise of feed water pressure increases the permeate water, the salinity intercepted on the membrane surface and the concentration polarization factor. When the feed water flow rate increases, the turbulence phenomenon intensifies, which causes the salt intercepted on the membrane surface to spread to the concentrated water. Consequently, the concentration polarization factor decreases.

In this study, SW30-2540 Dow membrane was selected according to the demand of 2 m$^3$ fresh water per day. The fitting formula of the concentration polarization factor is derived by ROSA9.0 on the basis of the influence parameters of the concentration polarization factor combined with the performance parameters provided by the membrane manufacturer.

$$\phi = 6.647 \times 10^{-8} \times P_f - 0.00326 \times \frac{Q_f}{3600 \times 24} + 0.855017 \tag{13}$$

where $P_f$ is the feed water pressure.

The water recovery ratio R indicates the ratio of the permeate water flow to the feed water flow:

$$R = \frac{Q_p}{Q_f} \times 100\% \tag{14}$$

The water produced by RO desalination is the feed water of MCDI. In MCDI desalination, Andelman et al. [17] deduced an effluent concentration models under constant voltage operation. The molar purification rate is

$$v_{mole} = \frac{\varepsilon V(t)}{zFR_{serise}} \exp\left(-\frac{t_{ad}}{R_{serise}C_{cap}}\right),$$  (15)

where $\varepsilon$ is the Coulomb efficiency. $V(t)$ is the on–load voltage. $z$ represents the averaged partial molar ionic valences of the feed water. $F$ is the Faraday constant. Then, $R_{serise}$ is the series resistance. $t$ is the adsorption time and $C_{cap}$ is the capacitance.

The residence time of the solution in the channel volume is determined by the channel volume ($V_s$) and the flow rate ($Q_{CDI}$), which can be expressed as:

$$t_{resistance} = \frac{V_s}{Q_{CDI}}$$  (16)

The number of ions adsorbed in the flow channel volume of the solution is deduced according to the integral of the molar purification rate to the time:

$$MOLES = \int_{t_n}^{t_n+t_{resistance}} v_{mole} dt$$  (17)

where $t_n$ is the time when the solution is in the flow channel.

Therefore, the concentration $C_s(t)$ in the flow channel at any time can be expressed as

$$C_s(t) = C_p - \frac{58.5 \times 10^6 MOLES}{V_s}$$  (18)

Given the existence of the dead zone, the mixing of the solution in the dead zone and the purified solution in the flow channel volume must be considered. The mixing ratio fraction ($t$) can be regarded as a function of time, which can be expressed as

$$fraction(t) = 1 - \exp\left(-\frac{tQ_{CDI}}{V_c}\right)$$  (19)

where $V_c$ is the dead zone volume.

The final effluent concentration in the adsorption stage is

$$C_{ad}(t) = C_p(1 - fraction(t)) + C_s(t)fraction(t)$$  (20)

On this basis, Janda [18] et al. deduced that the final specific effluent concentration calculation formula was

$$C_{ad}(t) = C_f - \frac{\mu}{V_s}[\exp(-\alpha t) - \exp(-\beta t)]$$  (21)

$$\mu = \frac{\varepsilon C_{cap} V}{zF}\left(1 - \exp\left(-\frac{V_s}{QR_{serise}C_{cap}}\right)\right)$$  (22)

$$\alpha = \frac{1}{R_{serise}C_{cap}}$$  (23)

$$\beta = \frac{1}{R_{serise}C_{cap}} + \frac{Q_{CDI}}{V_c}$$  (24)

where the series resistance includes the exchange membrane resistance, the solution resistance and the contact resistance between the collector and the electrode [19,20].

The series resistance can be expressed as

$$R_{serise} = sA_c + \frac{l}{\sigma A_c} \times 10^6 \tag{25}$$

where $s$ refers to all the area resistance except the ion area resistance. $\sigma$ is the solution conductivity. $l$ is the distance between the electrodes and $A_c$ is the carbon electrode area.

Other parameters set during the simulation process are shown in Table 1.

**Table 1.** Parameter design.

| Parameters | Value |
|---|---|
| Carbon electrode area density (g m$^{-2}$) [20] | 89 |
| Mass specific capacitance (F g$^{-1}$) [21] | 60 |
| Electrode plate spacing (mm) | 2 |
| Area resistance except the ion area resistance s ($\Omega$ cm$^{-2}$) [22] | $1.8 \times 10^{-4}$ |
| Charge voltage (V) | 1.2 |
| High-pressure pump efficiency | 80% |
| Seawater concentration (mg L$^{-1}$) | 35,000 |

The average effluent concentration in the adsorption process is

$$C_{avg} = \frac{\int_{t_1}^{t_2} C_{ad}(t)dt}{t_2 - t_1 + 1} \tag{26}$$

where $t_1$ is the start time of the adsorption and $t_2$ is the ending time of the adsorption.

The average effluent concentration in the desorption process is

$$C_{de} = \frac{C_p - C_{ad}\eta_{net}}{1 - \eta_{net}} \tag{27}$$

where $\eta_{net}$ is the MCDI water recovery rate, which refers to the ratio of the desalting water yield flow and the MCDI water inflow in an adsorption–desorption cycle.

We have verified through experiments that when the feed water concentration is less than 1000 μS cm$^{-1}$ and the water treatment rate (the difference between the average concentration of MCDI inlet and the yield level and the ratio of MCDI inlet concentration) is 60%, the MCDI water recovery rate $\eta_{net}$ can reach 90%. Therefore, the MCDI water recovery rate in this study is 0.9.

In the steady state, the influent concentration of RO is

$$C_f = \frac{C_{de}Q_p(1 - \eta_{net}) + C_{sea}\left[Q_f - Q_p(1 - \eta_{net})\right]}{Q_f} \tag{28}$$

where $C_{sea}$ is the seawater concentration.

### 3.2. Energy Consumption Calculation

In the RO desalting part, the consumption energy of the high-pressure pump is

$$E_{HP} = \frac{Q_f\left(P_f - P_0\right)}{\eta_{HP}} \tag{29}$$

where $P_0$ is the atmospheric pressure value. $\eta_{HP}$ represents the efficiency of the high-pressure pump.

The energy recovered by the energy recovery device is

$$E_{rev} = Q_c P_c \eta_{rev} \tag{30}$$

where $\eta_{rev}$ is the efficiency of the energy recovery device.

The specific energy consumption in RO production is:

$$E_{RO} = \frac{E_{HP} - E_{rev}}{1000 \times 3600 Q_p \eta_{net}} \tag{31}$$

In the MCDI desalination section, if the external resistance is ignored, the energy consumed in the charging process $W_{CDI}$ is theoretically equal to the ratio of the energy stored in the adsorption process to the charge efficiency [23]. As shown in Formula (32), $\eta_{charge}$ represents the charge efficiency.

The charge efficiency is defined as the ratio of the salt ion charge absorbed by the CDI device to the number of electron charges transferred between the electrodes during the adsorption–desorption cycle. In the case of high voltage, low influent concentration and the use of exchange membrane, the charge efficiency can be set to [17,24,25]

$$W_{CDI} = \frac{\frac{1}{2}C_{Cap}V^2}{\eta_{charge}} \tag{32}$$

The specific energy consumption of MCDI is

$$E_{CDI} = \frac{W_{CDI}}{3.6 \times 10^6 Q_{eff} t_{charge}} \tag{33}$$

where $Q_{eff}$ is the effluent flow rate in the adsorption process and $t_{charge}$ is the adsorption time in the adsorption–desorption process.

The total specific energy consumption of the system is

$$E = E_{RO} + E_{CDI} \tag{34}$$

### 3.3. Model Verification

In the part of the RO module, the performance parameters of the SW30-2540 membrane selected are shown in Table 2. The abovementioned simulation mathematical model is simulated and analyzed by MATLAB programming. The designed minimum pressure 3.5 MPa and maximum pressure 6.5 MPa, as well as the selected intermediate pressures of 4 MPa, 5 MPa and 6 MPa, are simulated when the feed water salt concentration is 32,000 mg L$^{-1}$ in the range of the feed water flow rate at 14–34 m$^3$ d$^{-1}$ and the feed water pressure at 3.5–6.5 MPa. The simulation results are compared with the ROSA9.0 simulation results. Figure 3 shows the comparison between the MATLAB simulation results and the ROSA9.0 simulation results under different pressures. The results show that the error is less than 6.4%, indicating that the simulation mathematical model and programming are credible.

**Table 2.** Performance parameters of SW30-2540 membrane.

| Characteristics | Value |
| --- | --- |
| Maximum feed water flow (m$^3$ d$^{-1}$) | 32.71 |
| Maximum water production (m$^3$ d$^{-1}$) | 2.23 |
| Maximum pressure (bar) | 68.95 |
| Permeability constant of pure water (m Pa$^{-1}$ s$^{-1}$) | $3.81 \times 10^{-12}$ |
| Solute transport parameter (m s$^{-1}$) | $5.815 \times 10^{-8}$ |
| Effective area of membrane (m$^2$) | 2.6 |

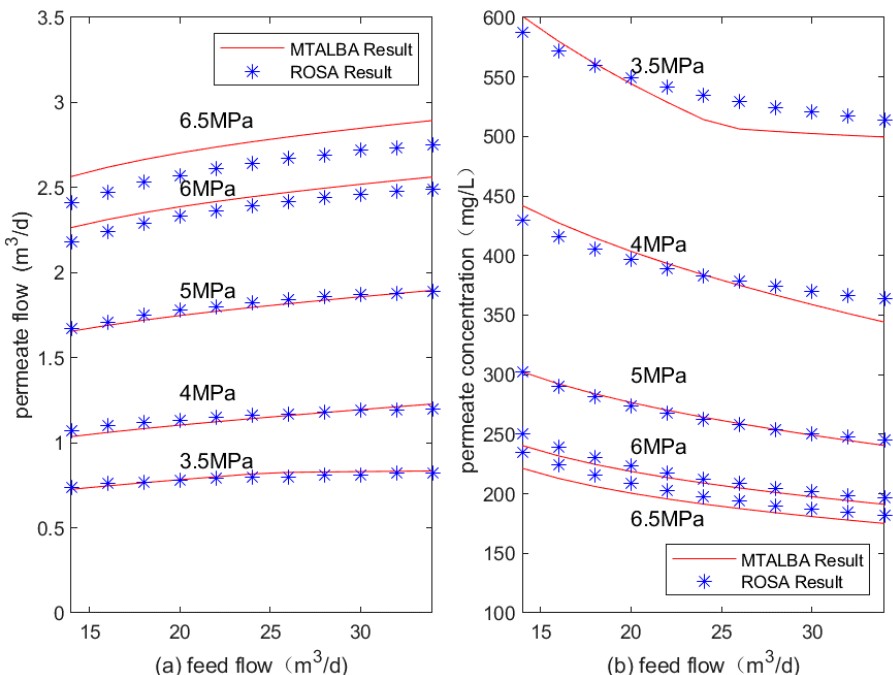

**Figure 3.** Comparison of MATLAB simulation results with ROSA9.0. (**a**) Relationship between feed water flow rate and produced water flow rate under different pressures; (**b**) relationship between feed water flow rate and produced water flow rate under different pressures.

In the MCDI desalination module, the simulation data of MATLAB programming are compared with the experimental results [26]. The operating parameters [27] are as follows: feed water flow rate of 60 mL min$^{-1}$; flow channel volume of 70 mL; dead zone volume of 100 mL; capacitance of 200 F; and feed water salt concentrations of 5 mmol, 10 mmol and 20 mmol. Figure 4 shows the comparison between the simulated effluent concentration curve of MATLAB and the experimental concentration curve. The results show that the effluent concentration curve of the established CDI mathematical model is in good agreement with the experimental data, and the results show that the error is less than 12%.

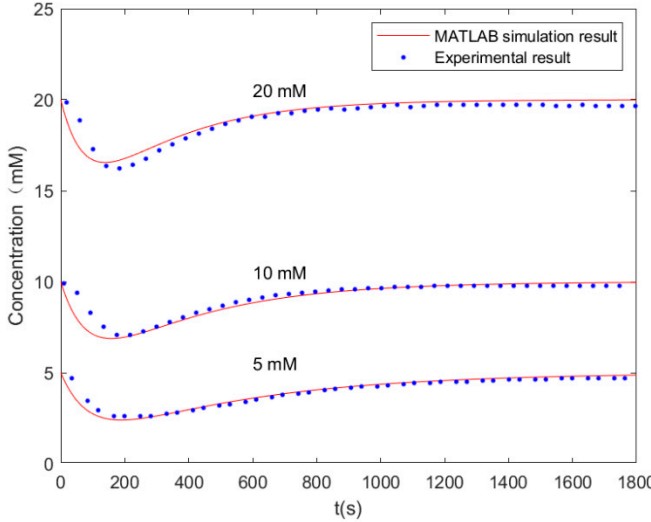

**Figure 4.** Comparison of MATLAB programming simulation results with experimental results.

## 4. Results and Discussion

The small-scale RO and MCDI coupling seawater desalination system with a daily production of 2 m³ fresh water and a salt concentration of less than 280 mg L⁻¹ is taken as the object of this study. The RO module chooses the SW30-2540 Dow membrane and takes the minimum energy consumption as the optimization goal. The parameters of the RO and MCDI coupling desalination system are matched and optimized. Then, the proposed system is compared with the pure RO desalination system. The feed water flow interval is (14 m³ d⁻¹, 34 m³ d⁻¹) and the feed water pressure interval is (3.5 MPa, 6.5 MPa) in the RO module. The number of channels interval is (30,100), and the length and width interval of the electrode plate interval is (5 cm, 20 cm) in the MCDI module.

The design schemes for the RO and MCDI coupling desalination system are the one-stage two-section RO+MCDI (1−2 RO+MCDI) and the one-stage three-section RO+MCDI (1−3 RO+MCDI). For comparison, the schemes for the pure RO desalination system are as follows: one-stage one-section RO (1−1 RO) and one-stage two-section RO (1−2 RO).

Considering the design schemes of the two-coupling desalination system, the influence of the RO module on the system performance is dominant. According to a given range of system constraints, the relationship among the different feed water flows, the different feed pressures, the specific energy consumption of RO, the water recovery rate of RO, the permeate water concentration of RO, and the daily water yield of the RO of the four desalination systems is given first, as shown in Figures 5–8. Then, the relationships among the number of different flow channels, flow channel area and specific energy consumption of the MCDI module within a given constraint range is shown in Figure 9.

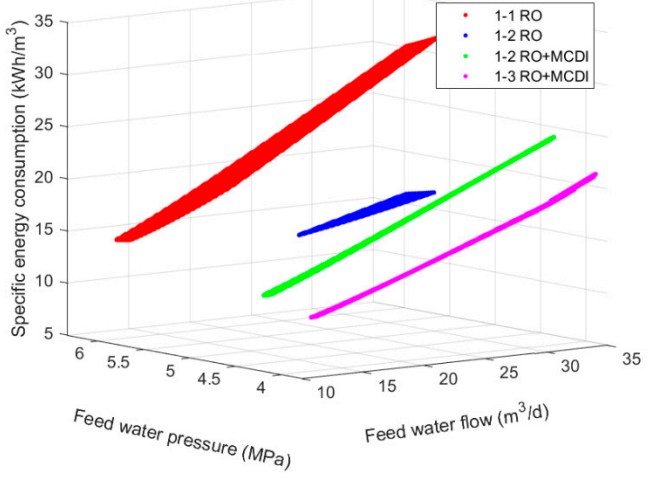

**Figure 5.** Relationship among feed water pressure, feed water flow and system-specific water energy consumption of RO under constraint condition of water production.

As seen in figures, in a pure RO desalination system, the lowest energy consumption condition of the 1−1 RO system is as follows. The feed water flow rate is 14 m³ d⁻¹, and the feed water pressure is 6.34 MPa. The specific energy consumption of RO is 13.63 kWh m⁻³. Moreover, the daily production of fresh water is 2.23 m³, the water recovery rate is 15.9%, and the average concentration of the produced water is 272.12 mg L⁻¹. In the minimum energy consumption of the 1−2 RO system, the feed water flow rate is 26.26 m³ d⁻¹. In addition, the feed water pressure is 5.98 MPa. The specific energy consumption of the RO is 13.1 kWh m⁻³. The daily production of fresh water is 4.09 m³, and the water recovery rate is 15.58%. Moreover, the average concentration of the produced water is 280 mg L⁻¹. The water recovery rates of the first-stage two-section RO and first-stage one-section RO are approximately the same. However, the water yield of the first-stage two-section RO is 1.83 times higher than that of the first-stage one-section RO. Moreover, the water supply pressure required is slightly lower, thus reducing the specific energy consumption.

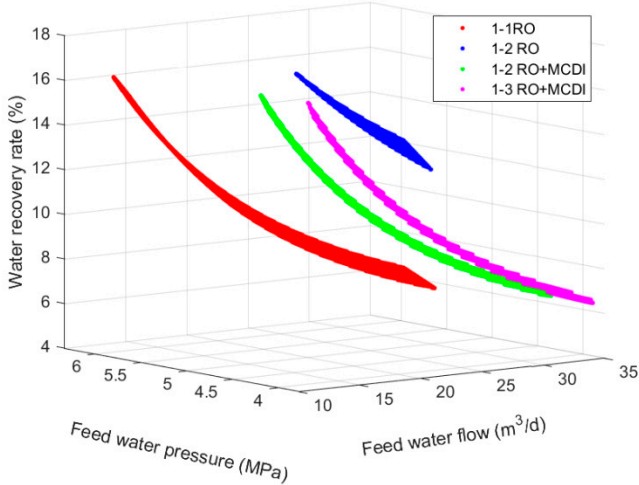

**Figure 6.** Relationship among feed water pressure, feed water flow rate and water recovery rate of RO under constraint condition of water production.

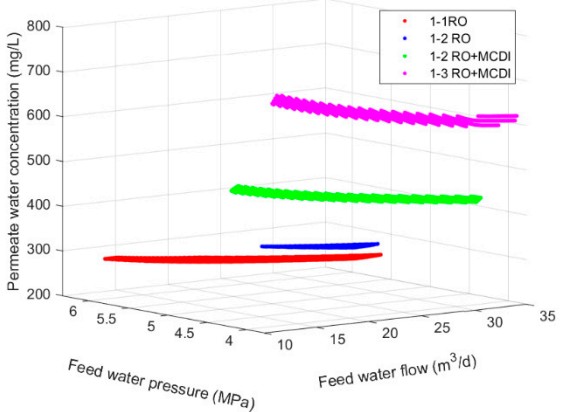

**Figure 7.** Relationship among feed water pressure, feed water flow rate and permeate water concentration of RO under constraint condition of water production.

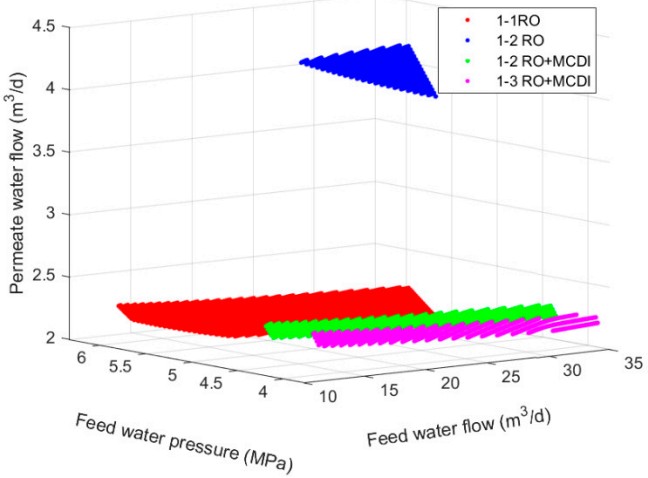

**Figure 8.** Relationship among feed water pressure, feed water flow rate and daily water yield of RO under constraint condition of water production.

In the 1−2 RO and MCDI coupling system, which is the lowest energy consumption condition of RO, the feed water flow rate is 14 $m^3$ $d^{-1}$. The feed water pressure is 4.72 MPa, and the specific energy

consumption of RO is 10.93 kWh m$^{-3}$. In addition, the daily production of fresh water is 2.28 m$^3$. The water recovery rate is 16.3%, and the average concentration of produced water is 477 mg L$^{-1}$. When the influent concentration of MCDI is 477 mg L$^{-1}$, the energy consumption of MCDI varies between 0.08 kWh m$^{-3}$ and 0.1 kWh m$^{-3}$ in the variable range of the number of channels and the area of electrodes. For the lowest specific energy consumption, when the number of flow channels is 32 and the length and width of flow channels are both 13.5 cm, the average concentration of the MCDI effluent is 280 mg L$^{-1}$. Moreover, the water production of the MCDI is 2.06 m$^3$ d$^{-1}$, and the energy consumption is 0.08 kWh m$^{-3}$.

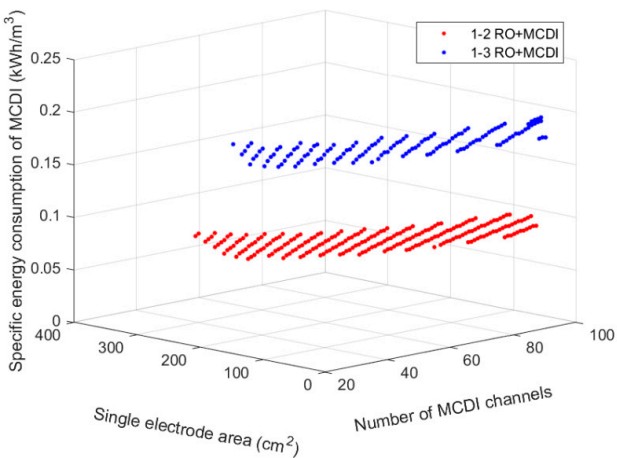

**Figure 9.** Relationship among number of channels, area of single electrode and specific energy consumption of MCDI.

For the 1−3 RO and MCDI coupling system, which is the lowest energy consumption condition of RO, the feed water flow rate is 14 m$^3$ d$^{-1}$, and the feed water pressure is 4.2 MPa. The specific energy consumption of RO is 9.66 kWh m$^{-3}$, and the daily production of fresh water is 2.29 m$^3$. Additionally, the water recovery rate is 16.4%, and the average concentration of the produced water is 689 mg L$^{-1}$. When the influent concentration of MCDI is 689 mg L$^{-1}$, the energy consumption of the MCDI varies between 0.17 kWh m$^{-3}$ and 0.19 kWh m$^{-3}$. For the lowest specific energy consumption, when the number of flow channels is 53 and the length and width of flow channels are both 9.5 cm, the average concentration of the MCDI effluent is 280 mg L$^{-1}$, the water production of MCDI is 2.05 m$^3$ d$^{-1}$ and the energy consumption is 0.17 kWh m$^{-3}$.

Further comparison of pure RO desalination can be observed. Although the yield water flow and yield water concentration meet the requirements, it is obtained under the premise of low flow and high-pressure feed water, which greatly increases the requirements for high pressure pumps. However, for the 1−2 RO+MCDI coupling desalination system, the RO effluent only needs to meet the water yield requirement. Moreover, it does not need to meet the water yield concentration requirement. The excess salt ions in the effluent water are removed through MCDI, thus greatly reducing the energy consumption and the feed water pressure before the RO membrane of the coupling desalination system.

If the 1−3 RO+MCDI coupling desalination system is adopted instead of the 1−2 RO+MCDI system, the salt concentration of the intermediate water yield of RO increases. However, the water supply pressure and energy consumption can be reduced even further for the whole system through MCDI desalination.

The simulation optimization results of the above four desalination methods are summarized in Table 3 and Figure 10, in which the specific energy consumption of each system is further shown when the energy of high-pressure concentrated brine in the RO desalination module is considered to be recovered by the high pressure energy recovery device. The energy recovery efficiency rates are 90%, 80% and 70%.

Table 3. Comparison of optimization results.

| Desalination Mode | Energy Recovery Efficiency of RO | Feed Water Pressure (MPa) | Feed Water Flow (m³ d⁻¹) | Number of MCDI Channels | Single Electrode Area (cm²) | Water Production Concentration (mg L⁻¹) | Water Production (m³d⁻¹) | Specific Energy Consumption of Desalination System (kWhm⁻³) |
|---|---|---|---|---|---|---|---|---|
| 1−1 RO | 0 | 6.34 | 14 | N/A | N/A | 272.12 | 2.23 | 13.63 |
|  | 90% |  |  |  |  |  |  | 5.54 |
|  | 80% |  |  |  |  |  |  | 6.44 |
|  | 70% |  |  |  |  |  |  | 7.34 |
| 1−2 RO | 0 | 5.98 | 26.26 | N/A | N/A | 280 | 4.09 | 13.1 |
|  | 90% |  |  |  |  |  |  | 5.46 |
|  | 80% |  |  |  |  |  |  | 6.3 |
|  | 70% |  |  |  |  |  |  | 7.16 |
| 1−2 RO+MCDI | 0 | 4.72 | 14 | 32 | 13.5 | 279.2 | 2.06 | 11 |
|  | 90% |  |  |  |  |  |  | 4.61 |
|  | 80% |  |  |  |  |  |  | 5.4 |
|  | 70% |  |  |  |  |  |  | 6.1 |
| 1−3 RO+MCDI | 0 | 4.2 | 14 | 53 | 9.5 | 280 | 2.05 | 9.83 |
|  | 90% |  |  |  |  |  |  | 4.24 |
|  | 80% |  |  |  |  |  |  | 4.86 |
|  | 70% |  |  |  |  |  |  | 5.48 |

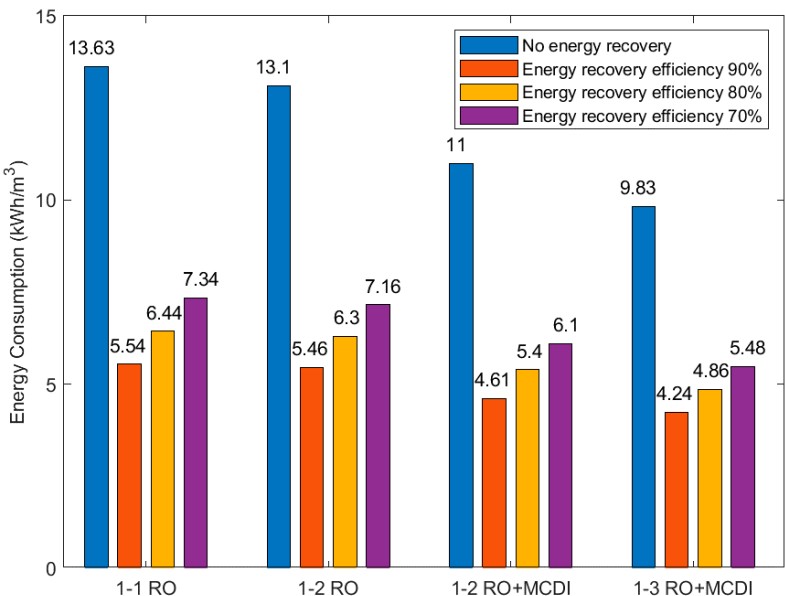

**Figure 10.** Comparison of energy consumption of four desalination methods.

## 5. Conclusions

This study proposes a seawater desalination system coupling RO and MCDI. Then, the RO simulation results are compared with the ROSA9.0 simulation results and the MCDI simulation results are compared with the experimental results by MATLAB. The relative errors are controlled within 6.4% and 12%, respectively. Afterward, the small-scale seawater desalination system with a daily production of two cubic meters of fresh water and a salt concentration of less than 280 mg L$^{-1}$ is taken as the object. Taking the minimum energy consumption of the system as the target, the two coupling design schemes of the seawater desalination system are compared and analyzed. In addition, the parameters are matched and optimized. The feed water pressure, feed water flow and specific energy consumption of the 1−1 RO system are 6.34 MPa, 14 m$^3$ d$^{-1}$ and 13.63 kWh m$^{-3}$, respectively. Meanwhile, the feed water pressure, feed water flow and specific energy consumption of the 1−2 RO system are 5.98 MPa, 26.26 m$^3$ d$^{-1}$ and 13.1 kWh m$^{-3}$, respectively. The feed water pressure, feed water flow and specific energy consumption of the 1−2 RO+MCDI system are 4.72 MPa, 14 m$^3$ d$^{-1}$ and 11 kWh m$^{-3}$, respectively. In the 1−3 RO+MCDI system, the feed water pressure, feed water flow and specific energy consumption of are 4.2 MPa, 14 m$^3$ d$^{-1}$ and 9.83 kWh m$^{-3}$, respectively.

In the range of pressure and flow that the Dow membrane can withstand, the performances of the one-stage two-section RO and MCDI coupling desalination system and the one-stage three-section RO and MCDI system are evidently better than that of the one-stage one-section and one-stage two-section pure RO desalination systems after parameter matching optimization. Our proposed system not only reduces specific energy consumption, but also lowers feed water pressure before RO membrane, which can reduce the selection standard of a high-pressure pump in a small seawater desalination plant. The coupling desalination method does not have a high requirement for the RO yield water concentration, and the part of RO desalination only needs to meet the water production. With the decrease of the feed water pressure, the production of a single membrane decreases, and the water concentration of a single membrane increases. Therefore, the desalination method of multi-section RO can be used to meet the requirements for water production.

The energy consumption of the one-stage three-section RO and MCDI coupling desalination system is lower than that of the one-stage two-section RO and MCDI coupling desalination system. The feed water pressure before the RO membrane is also lower, which can further reduce the standard of the high-pressure pump.

**Author Contributions:** Conceptualization, S.Y.; Formal analysis, M.J.; Methodology, S.Y.; Software, M.J.; Writing—original draft, M.J.; Writing—review & editing, S.Y. All authors have read and agreed to the published version of the manuscript.

**Funding:** This research received no external funding.

**Conflicts of Interest:** The authors declare no conflict of interest.

## Nomenclature

| | |
|---|---|
| $A$ | effective area of membrane (m$^2$) |
| $A_c$ | carbon electrode area (cm$^2$) |
| $A_w$ | permeation constant of pure water (m$^3$ m$^{-2}$Pa$^{-1}$s$^{-1}$) |
| $B_s$ | mass transfer coefficient of salt (kg m$^{-2}$s$^{-1}$) |
| $C_{ad}$ | effluent concentration in the adsorption stage of MCDI (mg L$^{-1}$) |
| $C_{avg}$ | average effluent concentration in the adsorption process (mg L$^{-1}$) |
| $C_B$ | average salt concentration on the feed side of RO (mg L$^{-1}$) |
| $C_c$ | the concentrated water concentration of RO (mg L$^{-1}$) |
| $C_{de}$ | average effluent concentration in the desorption process of MCDI (mg L$^{-1}$) |
| $C_f$ | feed water concentration of RO (mg L$^{-1}$) |
| $C_{sea}$ | seawater concentration (mg L$^{-1}$) |
| $C_m$ | concentration of membrane wall on the influent side of RO (mg L$^{-1}$) |
| $C_p$ | permeate water concentration of RO (mg L$^{-1}$) |
| $C$ | capacitance (F) |
| $E_{rev}$ | energy recovered by energy recovery device (W) |
| $E_{HP}$ | consumption energy of the high-pressure pump (W) |
| $E$ | total energy consumption of the system (kWh m$^{-3}$) |
| $E_{CDI}$ | specific energy consumption of MCDI (kWh m$^{-3}$) |
| $E_{RO}$ | specific energy consumption in RO production (kWh/m$^3$) |
| $F$ | Faraday constant (96,485 C mole$^{-1}$) |
| $j_w$ | water flux passing through the membrane (m$^3$ m$^{-2}$ s$^{-1}$) |
| $P_f$ | feed water pressure (Pa) |
| $P_c$ | concentrated water pressure (Pa) |
| $P_p$ | permeate water pressure (Pa) |
| $P_0$ | atmospheric pressure (Pa) |
| $Q_c$ | concentrated water flow rate (m$^3$ s$^{-1}$) |
| $Q_f$ | feed water flow rate (m$^3$ s$^{-1}$) |
| $Q_{eff}$ | effluent flow rate in the adsorption process (m$^3$ s$^{-1}$) |
| $Q_p$ | permeate water flow rate (m$^3$ s$^{-1}$) |
| $R$ | water recovery rate of RO |
| $R_{serise}$ | series resistance ($\Omega$) |
| $s$ | all the area resistance except the ion area resistance ($\Omega$ cm$^{-2}$) |
| $t$ | Celsius temperature (°C) |
| $t_{ad}$ | adsorption time (s) |
| $t_{charge}$ | adsorption time in the adsorption–desorption process (s) |
| $t_{resistance}$ | resistance time (s) |
| $V_c$ | the extra dead zone volume (mL) |
| $V_s$ | channel volume (mL) |
| $V(t)$ | on–load voltage(V) |
| $W_{CDI}$ | energy consumption in the process of capacitive deionization (J) |
| $z$ | averaged partial molar ionic valences of the feed water |

## Greek

| | |
|---|---|
| $\Delta P$ | difference of static pressure (Pa) |
| $\Delta \pi$ | osmotic pressure across the membrane (Pa) |
| $\pi_m$ | osmotic pressure on the membrane wall on the feed side (Pa) |
| $\pi_p$ | osmotic pressure of permeate water (Pa) |
| $\phi$ | concentration polarization factor |
| $v_{mole}$ | molar purification rate (mole s$^{-1}$) |
| $\sigma$ | solution conductivity ($\mu$S cm$^{-1}$) |
| $\eta_{charge}$ | charge efficiency |
| $\eta_{HP}$ | efficiency of the high-pressure pump |
| $\eta_{net}$ | water recovery rate of MCDI |

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
