# Peer review of "A Small RO and MCDI Coupled Seawater Desalination Plant and Its Performance Simulation Analysis and Optimization"

_processes, doi:10.3390/pr8080944_

Round 1

Reviewer 1 Report

Dear Yao et al.

Under energy, water, climate crisis, desalination is a very hot issue to sustain the clean water. Among various methods, RO is one of the innovative technologies with respeccing to the lowest energy consumption. However, the average energy consumption with seawater is around 3 kWh/m3 in industrial factories. Therefore, there are many efforts to jump up the current limits.

Unter this consideration, this paper deals with the hybrid system between RO and MCDI to minimize energy consumption. The authors suggest four types of systems on how to reduce energy consumption. Finally, they show the three stages RO and then MDCI system has the lowest energy consumption. Therefore, this paper is well suitable to be published in Processes journal, but there are some questions to make clear. 

  1. In abstract, "mining at producing 2 m3 ~~" --> Please double-check the grammar.
  2. In the introduction section, "For example, when the feed water ~~ the feed water flow rate is 100 mL" --> Please check the unit of flow rate. Need some time unit.
  3. In the introduction section, "The results show that under the condition of ~~ of RO-CCOCD system and RO-CCVCD system" --> in here what if RO-CCVCD? 
  4. In equation (4,5), What is 320 value? the Kelven temperature? (273) and to clarify the temperature, please "t" is replaced to uppercase "T:
  5. In equation (30), nHP --> nrev
  6. As you know well, the recovery rate in the seawater desalination plant is around 50%. However, the recovery rate in your experiments is around 15~16%. Could say if the recovery rate increases, the energy consumption would increase or decrease? 
  7. Also, the additional parts in RO desalination can increase OPEX and CAPEX. Thus, even though the energy consumption can be minimized using MCDI,  if OPEX or CAPEX increases, it must be re-considered in real plants. I suggest your options in the conclusion section.  

Author Response

Response to Reviewer 1 Comments

Point 1: In abstract, "amining at producing 2 m3 ~~" --> Please double-check the grammar.

Response 1:  In abstract, “aiming at producing 2 m3 fresh water per day with a salt concentration of less than 280mg L-1” is modified to “aiming at producing 2 m3 d-1 fresh water with a salt concentration of less than 280mg L-1”.

Point 2: In the introduction section, "For example, when the feed water ~~ the feed water flow rate is 100 mL" --> Please check the unit of flow rate. Need some time unit.

In the introduction section, “For example, when the feed water concentration is 35,000ppm, the feed water flow rate is 100 mL s-1 (8.64 m3 d-1)” (It has been revised in the manuscript).

Point 3: In the introduction section, "The results show that under the condition of ~~ of RO-CCOCD system and RO-CCVCD system" --> in here what if RO-CCVCD?

Response 3: This is my mistake to make RO-CVOCD into RO-CCVCD. I have revised it in the manuscript. In RO-CCOCD system, more ultrapure water can be produced, but the average concentration of ultrapure water is higher than that in RO-CVOCD system.

Point 4: In equation (4,5), What is 320 value? the Kelven temperature? (273) and to clarify the temperature, please "t" is replaced to uppercase "T:

Response 4: The equation (4, 5) refers to reference [16], and maybe 320 is the result of simplification, and has no physical meaning. The unit of temperature in reference [16] is ℃, which represents the temperature in degrees Celsius. I have replaced " t" to uppercase "T" in the manuscript.

Point 5: In equation (30),  -->  ,

Response 5: Thank you for your reminding. I have revised it in equation (30).

Point 6: As you know well, the recovery rate in the seawater desalination plant is around 50%. However, the recovery rate in your experiments is around 15~16%. Could say if the recovery rate increases, the energy consumption would increase or decrease?

Response 6: If the recovery rate increases, the specific energy consumption decreases. The water recovery rate of single reverse osmosis membrane is limited. If you want to achieve a 50% water recovery rate, you need about 7 reverse osmosis membranes in series, that is, one-stage seven-section system. The more reverse osmosis membranes are connected in series, the water recovery rate will be higher under the same feed water flow rate, but the permeate water concentration will be higher. In a small-scale reverse osmosis seawater desalination, there are few reverse osmosis membranes connected in series, so the water recovery rate is low.

Point 7: Also, the additional parts in RO desalination can increase OPEX and CAPEX. Thus, even though the energy consumption can be minimized using MCDI, if OPEX or CAPEX increases, it must be re-considered in real plants. I suggest your options in the conclusion section. 

Response 7: The RO reverse osmosis membrane needs to be replaced regularly due to its limited life. In addition, the water produced is slightly acidic which is very easy to corrosion pipes and valves and the residual ions are excessive which is harmful to human body. There are two ways to solve the water quality problem. One is to adopt RO desalination again, but the investment cost is higher and the energy consumption is higher. The second is to improve the water quality by adding drugs, which also has higher operation and maintenance costs. Since small seawater desalination plants are aimed at ordinary users rather than enterprise users, adding drugs to improve water quality is not suitable for ordinary users who lack relevant professional conditions and means. Because safety problems are likely to occur. When MCDI is used for secondary desalination, the initial investment cost and subsequent operation cost of the system are lower than that of RO for secondary desalination, and the water quality problem can be completely solved. Although MCDI secondary desalination will increase the initial investment cost and operation cost of the system, MCDI secondary desalination for small seawater desalination plants is not only easy to operate but also low in system operation and maintenance cost, compared with the improvement of water quality by adding drugs.

Reviewer 2 Report

Shouguang et al. Report on the capabilities of a RO and MCDI
combined plant for water desalination. They estimated the
effectiveness of their proposed system for a daily production of 2m3
of fresh water. The paper report very interesting results and all the
sections are properly structured. In addition, the most relevant
papers, even recently published are reported among the literature. In
the opinion of the reviewer, the results are well presented and all
possible interpretation are provided, as well as the conclusion are
convincing. Anyway, a deep and careful check of the language by a
native speaker is strongly suggested by the reviewer as well as the
use of appropriate terms and terminology.  Indeed, although the
scientific content of the paper is interesting, the readness is not
fluid.

Author Response

Thank you for your confirmation and reminding.An native English speakers have checked and revised the manuscript. Please see the revised version.

Reviewer 3 Report

В целом, можно отметить, что работа является актуальной и полной практического интереса, потому что исследования, направленные на решение одной из глобальных проблем в мире, нехватки чистой пресной воды, очень важны сегодня. Комментарии включают следующее:

  1. Прикладная часть исследования выглядит лучше фундаментальной. Это поднимает вопрос о фундаментальной составляющей этой работы.
  2. Было бы более понятно увидеть полные расшифровки сокращений, которые впервые появляются в тексте. чтобы облегчить восприятие статьи,
  3. Обязательно обращать внимание на формат ссылок. В этой версии статьи вместо ссылок появилась надпись «Ошибка! Справочный источник не найден. »

Author Response

Response to Reviewer 3 Comments

Point 1: Прикладная часть исследования выглядит лучше фундаментальной. Это поднимает вопрос о фундаментальной составляющей этой работы.

Response 1: I have supplemented the introduction in the manuscript, including the significance of small-scale seawater desalination and the values of some parameters.

Point 2: Было бы более понятно увидеть полные расшифровки сокращений, которые впервые появляются в тексте. чтобы облегчить восприятие статьи,

 Response 2: Thank you for your reminding. I have added abbreviations in front of the references to explain the meanings of abbreviations so that readers can better understand them.

Point 3: Обязательно обращать внимание на формат ссылок. В этой версии статьи вместо ссылок появилась надпись «Ошибка! Справочный источник не найден. »

Response 3: Think you for your reminding. I have added a link after the reference and the body part can be linked to the corresponding reference in the revised manuscript.

Reviewer 4 Report

Presented material is interesting but some changes are required.

  • there is mistake in references citation: instead numbers there is information about error
  • introduction supposed to be changed: more information from cited articles supposed to be added (concentration, energy consumptions, removal efficiencies - for example ,,product quality of CDI is equal to or better" - but what are the parameters? Please develop information from cited texts.
  • between units and numers supposed to be spaces
  • equations: what are the units of parameters used in equations - symbols are not enough
  • ,,Figure 4 shows the comparison" - please develop. mathematical model is in good agreement with the experiemntal data' is also not enough. It supposed to be say something about similarity or differences in %.
  • ,,As you can see IN the figures" instead from
  • ,,Water flow and yield water concentration can meet the requirements" - which requirements?
  • Conclusions supposed to be develop (similarity of model and real data, some most important, general values obtained during experiments, explanation of obtained results etc.)
  • Table 2 - I think mass transfer supposed to be express as kg s-1 instead m3 s-1?
  • English is proper but some little mistakes occured - I recommend to read and check text once again

Author Response

Response to Reviewer 4 Comments

 Point 1: there is mistake in references citation: instead numbers there is information about error

Response 1: Think you for your reminding. I have added a link after the reference and the body part can be linked to the corresponding reference in the revised manuscript.

Point 2: introduction supposed to be changed: more information from cited articles supposed to be added (concentration, energy consumptions, removal efficiencies - for example, product quality of CDI is equal to or better" - but what are the parameters? Please develop information from cited texts.

Response 2: I have supplemented the introduction in the manuscript, including the significance of small-scale seawater desalination and the values of some parameters.

Point 3: between units and numbers supposed to be spaces

Response 3: Thank you for your reminding. Spaces have been added between units and numbers.

Point 4: equations: what are the units of parameters used in equations - symbols are not enough

Response 4: Thank you for your reminding. All the formulas have been checked and the missing parameters have been added to the Nomenclature section.

Point 5: Figure 4 shows the comparison" - please develop. mathematical model is in good agreement with the experiemntal data' is also not enough. It supposed to be say something about similarity or differences in %.

Response 5: The revised manuscript added the following sentence: "and the results show that the error is less than 12%." after “The results show that the effluent concentration curve of the established CDI mathematical model is in good agreement with the experimental data.”

Point 6: As you can see IN the figures" instead from

Response 6: Thank you for your reminding. The manuscript has been revised.

Point 7: Water flow and yield water concentration can meet the requirements" - which requirements?

Response 7: Requirements include water yield flow greater than 2 tons per day and water yield concentration less than 280 mg/L.

Point 8: Conclusions supposed to be develop (similarity of model and real data, some most important, general values obtained during experiments, explanation of obtained results etc.)

Response 8: Thank you for your reminding. The part of conclusion has been revised in the manuscript.

Point 9: Table 2 - I think mass transfer supposed to be express as kg s-1 instead m3 s-1?

Response 9:  Mass transfer coefficient of salt in Table 2 represents . In order not to cause ambiguity, is named solute transport parameter instead of Mass transfer coefficient of salt. According to dimensional analysis, the unit of  is .

Point 10: English is proper but some little mistakes occured - I recommend to read and check text once again

Response 10: Thank you for your reminding. I have revised some mistake in the manuscript.

Round 2

Reviewer 4 Report

Manuscript has been improved and can be accepted in the present form.